

# A comprehensive oceanographic dataset of a subpolar, mid-latitude broad fjord: Fortune Bay, Newfoundland, Canada

Sebastien Donnet[1], Pascal Lazure[2], Andry Ratsimandresy[1], Guoqi Han[1]

[1] Fisheries and Oceans Canada, NAFC, 80 East White Hills Rd, St. John's NL, A1C 5X1, Canada
[2] Ifremer (French Research Institute for Exploitation of the Sea) Laboratoire d'Océanographie Physique et Spatiale, Centre Bretagne, ZI de la Pointe du Diable, CS 10070, 29280 Plouzané, France

*Correspondence to*: Sebastien Donnet (sebastien.donnet@dfo-mpo.gc.ca)

**Abstract.** While the dynamics of narrow fjords, i.e. narrow with respect to their internal Rossby radius, have been widely studied, it is only recently that interest sparked in studying the physics of broad fjords due to their importance in glacial ice melting (in Greenland, especially). Here, we present a comprehensive set of data collected in Fortune Bay, a broad, mid-latitude fjord located on the Northwest Atlantic shores. Aside from being wide (15- 25 km width) and deep (600 m at its deepest), Fortune Bay also has the characteristics of having steep slopes, weak tides and of being strongly stratified from spring to fall. Thus, and since strong along-shore winds also characterise the region, this system is prone to interesting dynamics, generally taking the form of transient upwelling and downwelling travelling along its shores, similar to processes encountered in broad fjords of higher latitudes. The dataset collected to study those dynamics consists of water column physical parameters (temperature, salinity, currents and water level) and atmospheric forcing (wind speed and direction, atmospheric pressure, air temperature and solar radiation) taken at several points around the fjord using oceanographic moorings and land-based stations. The program lasted 2 full years and achieved a good data return of 90%, providing a comprehensive dataset not only for Fortune Bay studies but also for the field of broad fjord studies. The data are available publically from the SEANOE repository (https://www.seanoe.org/data/00511/62314/; Donnet and Lazure, 2020).

## 1 Introduction

Fortune Bay is a broad fjord-like embayment located on the south coast of Newfoundland, a large island of the Northwest Atlantic (Figure 1). It is about 130 km long and 15-25 km wide, with a maximum depth of about 600 m. It is semi-enclosed from the shelf by a series of sills of about 100-120 m limiting depth. While situated in mid-latitudes (about 47°N) the marine climate of this region can be defined as subpolar due to the cooling effect of the cold, equatorward Labrador Current of arctic origin (Dunbar, 1951 and Dunbar, 1953). As a result, its waters are strongly stratified in summer (de Young, 1983, Donnet et al., 2018a) and its internal Rossby radius Ri is smaller than its width (Ri ~5-10 km), making it similar to large polar fjords in that regard (e.g. Cottier et al., 2010).

While dynamics of narrow fjords, i.e. narrow with respect to their internal Rossby radius, have been well studied, wide fjords dynamics are much less known (see Farmer and Freeland 1983, Inall and Gillibrand 2010 and Stigebrandt 2012 for reviews of





narrow fjords). Similarly to narrow fjords, and to any coastal areas, tides, winds, freshwater input and remote forcing (e.g. pycnocline and sea-level differences with shelf water) all play a role in the dynamics of broad fjords (e.g. see Cottier et al., 2010 for a review). However, having a width larger than their internal Rossby radius allows for each side to behave independently or have important 'wall-to-wall' effects (e.g. Cushman-Roisin et al., 1994, Jackson et al., 2018). In other words,

rotation induces cross-fjord variations, in stratification and/or flow, such as surface freshwater distribution, deep water flow and potential transient wind-induced upwelling/dowelling events (Cottier et al., 2010).

Due to their importance in climate change studies, interest in wide fjords such as those present in Greenland has grown in recent years (e.g. Straneo and Cenedese, 2015, Inall et al. 2015, Jackson et al. 2018). Nevertheless and due to their remoteness, available observational data for those important regions remains very scarce.

The first set of oceanographic studies dedicated to Fortune Bay was conducted by researchers and students of Memorial University of Newfoundland (MUN) from the late 1980s to the mid-1990s and focused on deep-water dynamics (de Young and Hay 1987, Hay and de Young 1989 and White and Hay 1994) as well as its lower trophic biology (Richard, 1987 and Richard and Haedrich, 1991). Later on and with the development of the aquaculture industry in the region, renewed interest led to new studies focusing on general geographic and oceanographic characteristics (Donnet et al., 2018b), hydrography

(Ratsimandresy et al., 2014, Donnet et al., 2018a), ocean currents (Ratsimandresy et al., 2019) as well as more specific dynamics induced by strong wind events (Salcedo and Ratsimandresy, 2013). Based on these latter studies, which focused on the inner part of the embayment, it became evident that a comprehensive and large-scale (i.e. bay scale) survey would be necessary to understand the dominant dynamics of this region.

To this end, an observation program took place from May 2015 to May 2017. The program was centered on the deployment

and recovery of oceanographic moorings, deployment and recovery of weather stations and tide gauges and on the collection of temperature and salinity profiles (Figure 1). The key objective and feature of this program was to measure the water column stratification and currents simultaneously at multiple sites, continuously through the four seasons. Along with the observations, a numerical model is being implemented to help understand the processes involved and to predict the transport of variables of interest (e.g. virus, sea-lice or organic material originating from or going into aquaculture farms). The main objective of this

paper is to report on the data products, describing the methods, limitations, estimated uncertainties and main results in the hope of being useful not only to further studies of the region but also more generally to the field of broad fjord dynamics studies. The dataset and its summary description are available at: https://www.seanoe.org/data/00511/62314/.

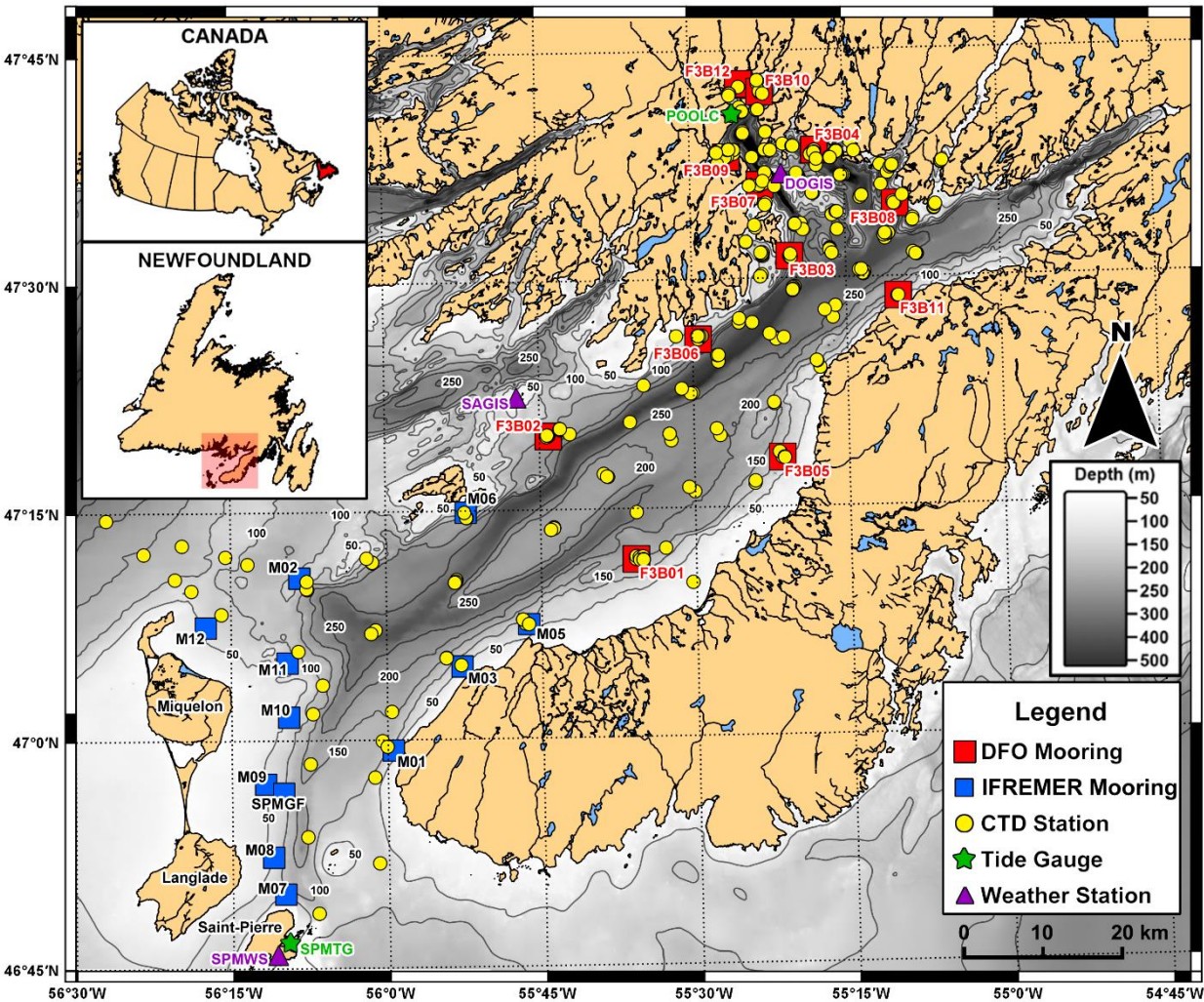

**Figure 1: Study area and summary of the observation program (May 2015 – May 2017)**

## 2 Material & methods

The observation program started in May 2015 with the deployment of 8 moorings at 4 sites (F3B01-04), a weather station (DOGIS) and a tide gauge (POOLC). The program lasted for two full years with maintenance trips occurring every 6 months. Thus, field operations occurred in May and November of each year for about 10-15 days each time. During each trip, additional measurements consisting of CTD (Conductivity, Temperature and Depth) profiles were collected and a separate trip was organised in August 2016 to get a better seasonal picture of the temperature and salinity field over the whole region. A small opportunistic survey, restricted to Belle Bay area, also occurred in June 2016 during the re-deployment of mooring F3B08.





The moorings consisted of a string of thermistors mounted with a couple of CTDs (one within each main hydrographic layer) and one (year 2) or two (year 1) ADCPs (Figure 2). The setup changed from year 1 (May 2015 – May 2016) to year 2 (May 2016 – May 2017) by merging the originally separated ADCPs and thermistor-CTDs lines (by about 100-150 m); thereby

doubling the amount of main sites being monitored from 4 (F3B01-04) to 8 (F3B01-08). Two other moorings lines were added, F3B09&10 and F3B11&12 for leg 3 (May-November 2016) and 4 (November 2016 - May 2017), respectively, to further increase the spatial resolution. With the exception of leg 1 (May-November 2015) F3B01&02, all moorings were sub-surface, taut-line type. A surface spar buoy was used on F3B01&02 during leg 1 in an attempt of measuring near-surface conditions and as a deterrent to fishing and shipping activities. The experiment was, however, unsuccessful with the loss of both surface

buoys after about 5 months deployment due to wave action wearing the mooring lines. Of those surface measurements, only one CTD dataset was partially recovered (RBR#60134 from F3B01, found on the shore with its spar buoy). Two main types of mooring were used during year 1 (Figure 2), an "ADCP" type having a set of two upward-looking ADCP separated by a string of thermistors and a "CTD" type consisting of two CTDs separated by a string of thermistors. The CTD type was declined in two version for leg 1: surface (F3B01&02) and sub-surface (F3B03&04). For leg 2 (November 2015 - May 2016),

only the subsurface version was retained; adding a 9 m rope on the top part of F3B01&02. In year 2 (leg 3 and leg 4), the "CTD" mooring design of year 1 was used as a base and equipped with an ADCP on the bottom part (around 80 m) to combine water stratification with ocean currents measures for most of the sites (F3B01-08). On F3B09 and 10, a simpler design was used due to the shallower depth of the sites, a need for less buoyancy and limitation on available hardware. To minimize drag we used ¼" dyneema ropes and OpenSeas SUBS buoys (Hamilton et al., 1997). CTDs were mounted in stainless-steal cages

for protection and thermistors were simply attached on the rope using cable ties and electrical tape. In most cases, acoustic releases were mounted in tandem for redundancy. Cooperation with our partner IFREMER (Institut Français de Recherche pour l'Exploitation de la MER) resulted in other sites being equipped with either bottom mounted thermistor (M01-10 Mastodon, Lazure et al., 2015) or ADCP moorings (SPMGF) during some of the legs (Leg 3 for M01-10 and all along for SPMGF).



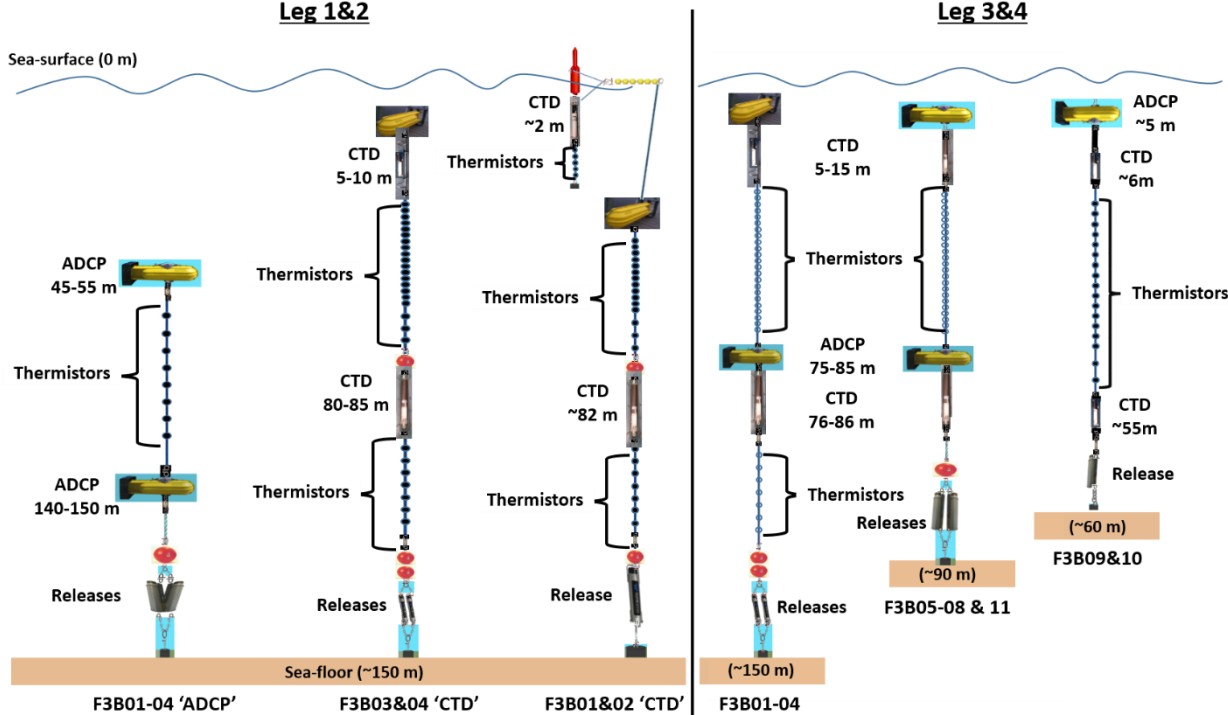


**Figure 2: DFO taut line moorings, leg1&2 (left), leg 3&4 (right). In leg 2, F3B01&2 'CTD' lines were converted to F3B03&04's design due to failures of the sea-surface part. A 110 m rope, without thermistors, was added under the bottom CTD of F3B08 to be deployed**

The other fixed (and long-term) structures were land-based and consisted of a weather station measuring wind speed and

direction (at 2 m and 10 m height above ground) as well as barometric pressure, air temperature, solar radiation (Qs) and

Photosynthetically Active Radiation (PAR) and of a tide gauge measuring the sea level and sea surface temperature. Two

weather station structures were installed on a small, barren island (DOGIS, see Figure 1 for location and Appendix A for an

illustration): a 2 m height tripod on which was mounted a wind sensor, a barometer, a temperature sensor, a pyranometer (i.e

solar radiation sensor) and PAR sensor and a 10 m mast on which was mounted another wind sensor. The tide gauge was

installed on a wooden wharf at the head of the bay (POOLC, see Figure 1 for location and Appendix A for an illustration) and

equipped with a vented pressure sensor mounted below chart datum in a black PVC tube to limit biofouling. These atmospheric

and tide observations completed existing sites equipped by other agencies (Figure 1): Sagona Island (SAGIS) weather station

(Environment and Climate Change Canada), St Pierre airport weather station (Meteo France) SPMWS, and St Pierre harbour

tide gauge (Service hydrographique et océanographique de la Marine, France) SPMTG.




## 2.1 Instruments used

A variety of instruments was used during this program, selected for their proven and common use in the field of physical oceanography and atmospheric science. All our ADCPs were WorkHorse models (WH) from Teledyne RDI (T-RDI); most of them were 300 kHz type though a few 600 kHz and one 1200 kHz (WH300, WH600 and WH1200, respectively) were also used during the second year. Most of the CTDs were SeaBird Electronic (SBE) instruments, model 19 ('SeaCAT' manufactured in the 90s) with a few 37 ('microCAT' manufactured in the 2000s). A few RBR concerto CTDs as well as XR420 Temperature-Depth-Dissolved Oxygen instruments were used on some legs (typically as backup and/or complementary observations). All the thermistors used were disposable Onset HOBO TidBiTs (UTBI) and a few Onset HOBO U20 thermistors equipped with a pressure sensor were also used to complement the UTBI and provide additional depth information of the mooring line. Gill windsonic sensors were used on our weather stations to measure wind speed and direction and were plugged to an Onset HOBO U30 logger on the 2 m tripod and to a Sutron SatLink2 logger with real-time transmit capability on the 10 m mast. An Onset Smart Barometric Pressure Sensor barometer (model S-BPA-CM10), an Onset 12-Bit Temperature Smart Sensor air temperature sensor (model S-TMB-M002), an Onset Silicon Pyranometer Smart Sensor (S-LIB-M003) and an Onset PAR Smart Sensor (S-LIA-M003) were also mounted on the 2 m tripod. A Sutron submersible pressure transducer (model 56-114) was used for the tide gauge; plugged to a Sutron SatLink2 logger with real-time transmit capability. Characteristics and specifications of all the sensors used are provided in Table 1.

## 2.2 Instruments limitations and uncertainties

Due to their difference in memory and battery capacity, sampling strategy (i.e. interval) differed from one instrument to another. All the ADCPs were set to sample every 30 min during leg 1-3. For leg 4, a higher sampling rate of 5 min was chosen to increase temporal resolution on moorings F3B03-12. In year 1, ADCPs were setup in 'burst mode', that is sampling for a smaller amount of time than their sampling interval (7.5 min vs. 30 min) to avoid possible cross-talk interference since two instruments of the same frequency were used on the same line. In year 2, all the ADCPs were sampling evenly (i.e. continuously) along the sampling average period. Higher vertical resolution (1 m cell) and broadband mode were used during leg1&2 for the near-surface units while lower vertical resolution (3m cell) and narrowband mode was used for the near-bottom units to maximise range. Overall, a reduction of about 30% in profile range from the manufacturer specifications was found due to the clarity of the water (i.e. low backscattering volume conditions). Based on first year results, the sampling strategy was re-thought to increase the horizontal sampling in year 2 (8 or more sites vs. 4) while keeping vertical profiling of the stratified part of the water column. i.e. from about 10 m to 80 m depth. Cell size was increased from 2 m (leg 3) to 3 m (leg 4) to prevent from losing range during very clear water conditions usually observed in winter. Narrowband mode was used for all our units during year 2 for the same reason.

SBE & RBR CTDs were all set to sample at 20 min interval while the XR420 were set at 1 min interval during leg 2 and 20 min during leg 3&4. The UTBI were set to 10 min interval along with the SUTRON weather station and tide gauge (with a 1





min internal average for the SUTRON). The U30 weather station was initially setup with a 30 min interval with no averaging during leg 1&2 and then adjusted to a 10 min interval, 30 samples averaged, during leg 3&4.


**Table 1: instrumentation used, sampling setup and stated uncertainty (i.e. noise) based on manufacturer specification and sampling setup. "top" and "bottom" refers to ADCP position on the mooring line during Leg1&2 (about 50 m vs. 145 m depth, respectively).**

| Instrument | Sampling Interval (# sample averaged) | Uncertainty |
|---|---|---|
| T-RDI ADCPs | 30 min (120) – Leg1&2 top<br>30 min (60) – Leg1&2 bottom<br>30 min (200) – Leg 3<br>30 min (200) – Leg 4 F3B01-02<br>5 min (33) – Leg 4 F3B03-12 | 0.7 - 1.7 cm/s<br>0.03 - 0.07 ºC<br>1.4 - 3.5 cm |
| SBE19 CTDs | 20 min (1) | 0.01 ºC<br>0.02 (Salinity)<br>10 - 30 cm (unit dependant) |
| SBE37 CTDs | 20 min (1) | 0.002 ºC<br>0.006 (Salinity)<br>2% (DO)<br>1 cm |
| RBR Concertos | 20 min (60) | 0.0003 ºC<br>0.0008 (Salinity)<br>1.3 cm |
| RBR XR420 | 1 min (1) – Leg 2<br>20 min (60) – Leg 3&4 | 0.0004 - 0.002 ºC<br>2% (DO)<br>4.6 - 25 cm |
| HOBO UTBI | 10 min (1) | 0.21 ºC |
| HOBO U20 | 15 min (1) | 0.44 ºC<br>12 cm |
| Mastodon | 1 min (1) | 0.1 °C |
| HOBO U30 | 30 min (1) – Leg 1&2<br>10 min (30) – Leg 3&4 | 0.4 - 2 % wind speed<br>0.5 - 3 º wind direction |



| | | 0.9 - 5 mbar (atmospheric pressure) |
| --- | --- | --- |
| | | 0.04 - 0.2 ºC (air temperature) |
| | | 12 - 64 W/m$^2$ (solar radiation) |
| | | 40 - 125 umol/m$^2$/s (PAR) |
| HOBO U20 (weather station) | 15 min (1) | 0.44 ºC (air temperature) |
| | | 12 mbar (atmospheric pressure) |
| SUTRON weather station | 10 min (60) | 0.3 % wind speed |
| | | 0.9 º wind direction |
| SUTRON tide gauge | 10 min (1) | 0.3 cm |
| | | 1 ºC |

**2.3 ADCP backscatter processing**

To provide with some added value, the ADCP backscatter data were processed to convert the raw Returned Signal Strength Indicator (RSSI, E in equation below), a measure of acoustic pressure received by the transducers, to a corrected backscatter volume Sv, proportional to the amount (i.e. volume) of particles present in the water column. The method used to do the correction is an updated version of the popular Deines' method (Deines, 1999) published by Mullison (2017) and summarized by this equation (Mullison 2017, eq. 3):


$$Sv = C + 10\log((Tx+273.16)R^2) - 10\log(L_{DBM}) - 10\log(P_{DBM}) + 2aR + 10\log(10^{Kc(E-Er)/10} - 1)$$

Factory calibrated values of Kc (count to decibel factor) and Er (noise floor) were used to solve this equation, along with the temperature measured at the transducer head by the instrument (Tx). Transducer temperature and salinity value selected during

instrument setup (ES command; 32 in our deployments) were used to calculate the water absorption (a) along the range (from transducer) R; thereby implying homogeneous water conditions. The transmit pulse length LDBM was calculated using bin size (1-3 m) and beam angle (20º) values. Default values of constant C and transmit power PDBW provided by Mullison (2017, table 2) were used.

Overall, a combined uncertainty of 5 dB is estimated due to the assumption of water column homogeneity (constant absorption,

0.5 dB maximum error in summer toward the surface), the assumption of constant power source (±3 dB with alkaline batteries, reducing in transmit power with time) affecting Leg1&2 ADCPs more than Leg3&4 ADCPs which were using lithium batteries (featuring a quasi-constant transmit power all along a deployment) and inherent transducer linearity uncertainties (±1.5 dB according to Deines, 1999). This uncertainty is relatively small in comparison to the 55 dB average range (-90 to -35) observed along the program, i.e. less than 10%, though not negligible.



## 3 Results

### 3.1 Data return and coverage

In total, 40 ADCP timeseries, 60 CTD/TD-DO timeseries (33 SBE19, 16 SBE37, 6 RBR XR420 and 5 RBR concerto), 35 UTBI string series, 13 U20 timeseries, 11 Mastodon thermistor series, 16 weather station series (6 U30, 6 SUTRON and 4 U20) and 4 tide gauge series were collected (Appendix B, Table B-1 to Table B-6). Taken together, these timeseries amount to about 28715 record-days (about 79 years).

Percent coverage presented in Appendix B (Table B-1 to Table B-6) are calculated based on the recovered instruments and data only. Lost instrument or instrument from which no data could be recovered are not presented. In total: 1 CTD was lost (RBR#60135 on F3B02, Leg 1), 1 CTDs flooded without any possible data recovery (SBE19#1310 on F3B04 leg 2) and 14 UTBI were lost (5 on F3B01-CTD leg 1, 5 on F3B02-CTD leg 1, 1 on F3B02-ADCP leg 2, 2 on F3B02-CTD leg 2 and 1 on F3B01 leg 3). Outside the failure of F3B01&02 top mooring part (10 units lost at once), UTBI were typically lost during grappling operations when releases could not be triggered. There were also one failed deployment attempt at F3B08 during Leg 3 (acoustic release failure in May 2016) which was successfully re-deployed in June 2016 but resulted in about 30 days of observation time lost (from early May to early June).

Percent coverages were calculated based on the good data recovery of current speed, current direction, current vertical speed and water column backscatter volume for the ADCPs; temperature, salinity and depth for the CTDs; temperature, dissolved oxygen and depth for the RBR TD-DO sensors; temperature and depth (or atmospheric pressure when used as a weather station sensor) for the U20s; temperature for the UTBIs; depth and temperature for the SUTRON tide gauge; wind speed and direction for the SUTRON weather station and wind speed, wind direction, atmospheric pressure, air temperature, PAR and solar radiation for the U30 weather station. For each instrument, the percent coverage represents the useable data covering the expected periods of observation; for a multiple parameters instrument (as listed above) the percent coverage was calculated for each parameter and then averaged per instrument.

Overall, the coverage is about 93% without considering instrument lossest and about 91% when considering the losses. Illustrations of the data coverage is given in Figure 3 and Figure 4 as Gantt charts.

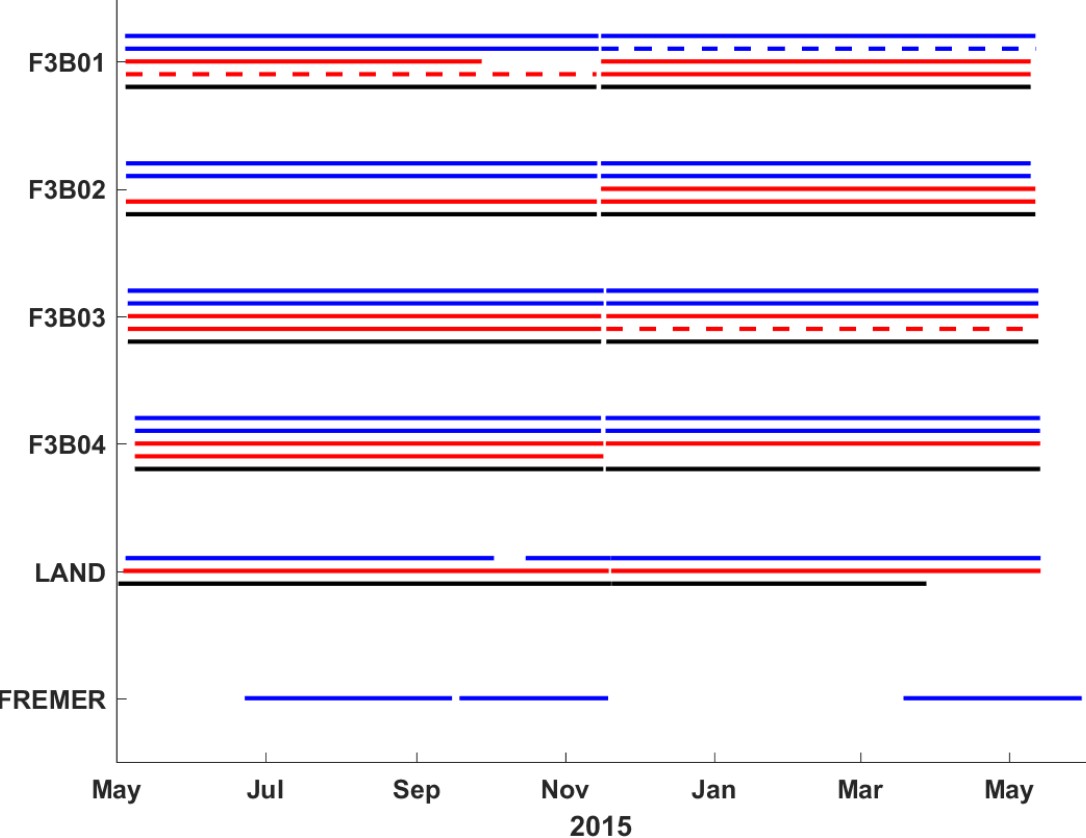

**Figure 3: Data return, leg 1&2. For the moorings F3B01-04, ADCPs are in blue, CTDs are in red and UBTIs are in black. Top line correspond to the shallowest unit. For the land-based stations (LAND), DOGIS SUTRON weather station is in blue, DOGIS U30 weather station is in red and POOLC SUTRON tide gauge is in black. IFREMER ADCP data (SPMGF) is in blue. Dash lines represent partial data recovery (e.g. ADCP tilted, CTD having no or partial salinity return).**


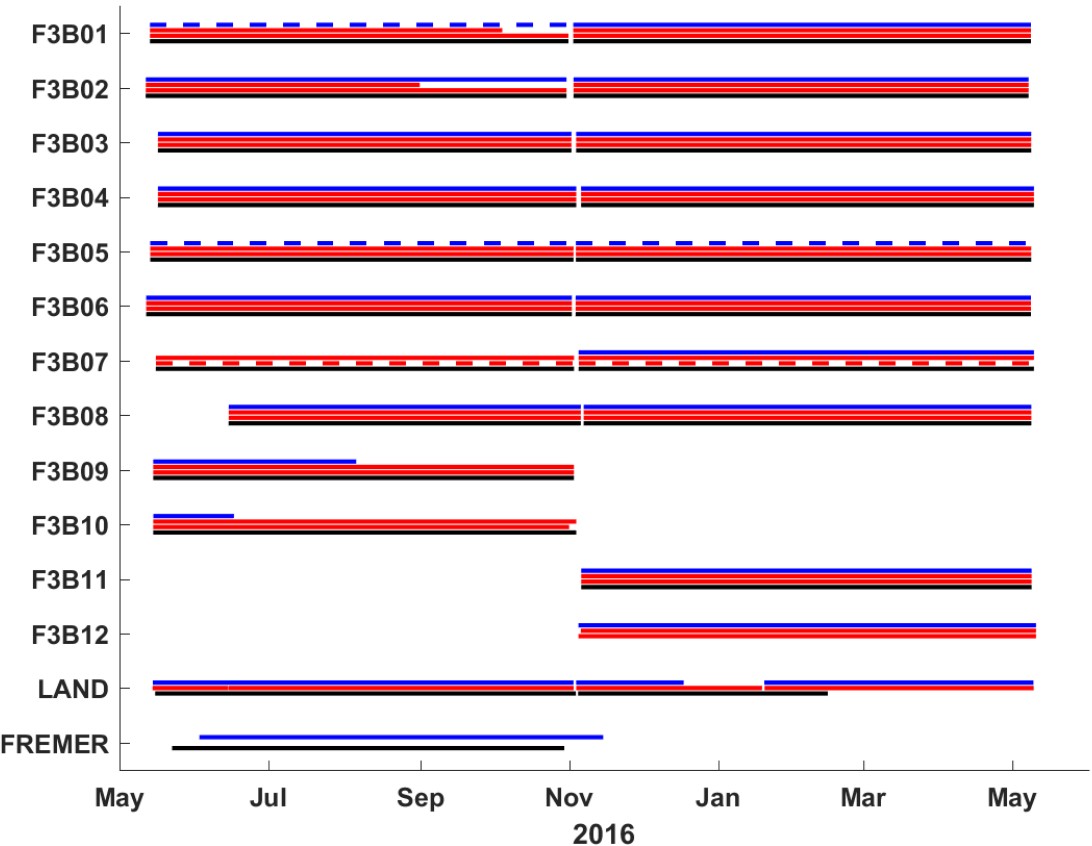

**Figure 4: Data return leg 3&4. For the moorings F3B01-12, ADCPs are in blue, CTDs are in red and UBTIs are in black. Top line correspond to the shallowest unit. For the land-based stations (LAND), DOGIS SUTRON weather station is in blue, DOGIS U30 weather station is in red and POOLC SUTRON tide gauge is in black. IFREMER ADCP (SPMGF) and MASTODON (M01-12) data are in blue and black, respectively. Dash lines represent partial data recovery (e.g. ADCP tilted, CTD having no or partial salinity return).**

## 3.2 Program validation

The primary objective of this observation program was to collect a robust baseline for studying the main physical processes affecting Fortune Bay. In particular, upwelling and downwelling propagation associated with strong currents along the shoreline was thought to be important features based on previous work done locally (Salcedo and Ratsimandresy, 2013, Donnet et al. 2018a) and in other embayments of the region (Yao 1986, de Young et al. 1993, Davidson et al. 2001, Ma et al. 2012). Hence, Fortune Bay's strong seasonal stratification, steep slopes, weak tides, strong along-shore winds and large width all indicated a potential for such 'coastally trapped' processes to occur.

The observation program was therefore designed to measure water vertical stratification and currents as well as forcing (i.e. wind and sea-level) over timescale of tens of minutes to a year-long and taken at as many points as possible along the coast,



within one internal Rossby radius, to follow potential disturbances travelling around the bay. Such features were indeed
observed and an example of which is presented in Figure 5. The study of those features, including their generation, propagation,
scale and importance on particles advection and water renewal, key aspects in studying the effect of aquaculture on the
environment, will be the focus of future publications.

## 4. Discussion

The uncertainty estimates presented in Table 1 are based on the instrument specifications and sampling strategy. That is, they
represent the expected short-term (i.e. noise) fluctuation around the true measure and assume a perfectly calibrated instrument,
i.e. no bias. Laboratory testing and in-situ performance checks were performed to further assess these estimates and correct
for eventual bias. Laboratory testing were performed in a 3 m depth seawater tank (for the CTDs, mainly) to check temperature,
salinity and depth measurements and a stable temperature water-bath was used to check temperature measurements (for the
UTBI, mainly). In-situ checks were obtained using CTD casts taken just after deployment and right before recovery of the
moorings and by cross-checking/comparing each instrument from the same mooring line (e.g. pressure measurements). The
main biases found were with the pressure sensors of the moored SeaBird Electronic model 19 (SBE19) instruments, which
could be as large as 6 dbar (~6 m of water depth). A combination of tank test results and in-situ check using the ADCP (and
other instruments when available) and mooring line length were used to determine these pressure biases. Both pressure sensor
data and corrected backscatter data (i.e. converted to volumetric backscatter values Sv in dB) were used to determine the in-
situ depth of the ADCP (i.e. the distance to water surface when using backscatter values) which was then used to crosscheck
the depth of the CTDs along the line. Biases that could not be determined with reasonable certainty resulted in discarding the
data. Except for leg 1, each moored CTD was sent to the manufacturer for calibration prior to deployment. For leg 1, only
laboratory tests could be done. All the thermistor used were new, i.e. bought for this program, and the ADCPs were 3-7 years
old. The CTD profilers used were sent to the manufacturer for calibration on a yearly basis with a 3-year rotation scheme. i.e.
3 CTD profilers were available for this program and 1 profiler was sent per year to be used as a reference for the other 2 in
laboratory and in-situ calibration/performance checks. Overall, it is estimated that the absolute depth of each instrument is
known to the nearest meter, that the temperature measured are accurate at ±0.2°C (UTBIs) or less (CTDs), that the salinity
measures are within 0.1 (moored CTD) or less (CTD profiles) from the true value and that the current speed is at ±2cm/s or
better. Note that if further averaging were to be done on those original timeseries, the uncertainty would go down by the square
root of the number of samples taken per sampling average except for the ±1m uncertainties on depth which can be seen as an
unknown bias.

**Figure 5: Fortune Bay water column thermal stratification and currents from July 1-20, 2016 showing several upwelling and downwelling events associated with strong currents 'pulses' travelling around the bay, i.e. from moorings F3B01 to F3B06.**





## 4.2 Data limitations and issues

The program suffered from some instrument failures. ADCPs from F3B09&10 during leg 3 suffered from a battery failure (F3B09) and from a memory card failure (F3B10) resulting in data coverage of only 48% (slightly less than 3 months) and 17% (about 1 month), respectively. Two SBE19 CTDs (#1310 and #1312) got their electronic casing flooded resulting in a complete loss of data on F3B04, leg 2 (SBE19#1310) and in partial losses (small leak) on F3B03, leg 2, and on F3B07, leg 3&4 (SBE#1312, temperature and salinity data corrupted). The DOGIS SUTRON weather station suffered from a solar panel failure during leg 1 resulting in a reduced coverage of 92% (a loss of about 12 days) and from a wind sensor failure during leg 3 (%56 coverage, a loss of about 33 days). DOGIS U30 weather station suffered from barometer issues during leg 1, 2 and 3 reducing coverages to 96%, 83% and 87%. The POOLC tide gauge also suffered from sensor failures, during both leg 2 and leg 4 of the program, resulting in reduced coverages of 73% (47 days lost) and 54% (86 days lost) and no coverage during the late winter-early spring seasons.

The program also suffered from some human errors and practical difficulties. Notably, several tangling of mooring ropes has resulted in excessive vertical tilt orientation of the upward-looking ADCP on moorings F3B01, 05 and 07 of leg 3 and on F3B05 leg 4, corrupting the data (see details below). In the case of leg 3 (F3B01, 05 and 07), problematic deployments in which the mooring line was not properly kept tight prior to releasing the anchor likely played a role. In the case of leg 4 (F3B05) it is less obvious since a stricter mooring deployment procedure was then in place and that field work records do not indicate any wrong doing. The use of SUBS buoys, though improving mooring drag and potential 'knock-down' from strong currents, increases deployment difficulty when they are placed in the middle of a mooring line (as opposed to the top of a line) since they have a natural tendency of orienting themselves in the flow; thus, to have the rope close to their back fin when sinking downward, thereby increasing chances of being tangled. It should be noted that one case of tangling/excessive ADCP vertical tilt occurred to the bottom ADCP of F3B01 during leg 2, after about 3 weeks of deployment (see below for details and Figure 7); thus not likely due to a deployment issue. Fishing activities may have caused it though no evidence of it could be found by looking at the data (e.g. rise and fall of the mooring); the data sampling frequency of the ADCP (30 min) prevents from a fine examination though no obvious evidence of mooring movement could be seen with the higher frequency UBTI records either (10 min) and fishing activities during that time of the year (December) is not very likely. The issue occurred during a strong current event, indicating that strong current shear could potentially be an actor.

## 4.3 QA/QC and data processing methods

Data were processed and quality checked similarly for all the instruments, that is:

1) raw data were first converted to the most convenient format known/available to the authors

2) time stamp and all variables of interest were extracted from the raw data and meta-data were associated to the dataset (i.e. station ID, geographical coordinates, deployment and recovery date and time and instrumentation ID)

3) using deployment and recovery time, 'out-of-water' data were removed



4) clock-drift and depth offset were assessed and corrected using concurrent data available on the same line. ADCPs were the most often used as a reference since their pressure sensor were systematically 'zeroed' prior to deploy and that their clock did not drift more than a few minutes per deployment. U20s and RBRs were usually used as a secondary reference or as a primary one when no ADCP were available on the same line (e.g. in leg 1&2). SBE19 units were the most affected by clock-drift and depth offset. A few units (SBE19 as well as UTBI) were also found to have been setup in local time instead of UTC, mistakenly.

5) 'out-of-range' data were removed using automatic filters following the criteria shown in Table 2. ADCP criteria was largely based on the manufacturer recommendations with current speed less than 0 (bad values are actually logged as -32768; see T-RDI documentation #P/N 957-6156-00, p147), Percent Good (PG) less than 25 (T-RDI documentation #P/N 957-6156-00, p150) and instrument tilt over 15 degree from the vertical (T-RDI documentation #P/N 957-6150-00, p17) used to remove bad data. Instrument vertical tilt was calculated using the pitch and roll records (see below for details). In addition to those data quality filters, a 'surface rejection' filter was applied as a percentage of the range to sea-surface (or sea-bottom for the downward looking F3B09&10) usually equal to 10% (i.e. a little higher than the theoretical 6% stated for 20º beam angle ADCPs, T-RDI documentation #P/N 951-6069-00 p38). Trial and error was performed for this latter filter by examining the velocity, backscatter and correlation profiles of each of the timeseries. In the case of severely tilted instrument (details below) up to 30% of the range needed to be removed. Speed, PG and surface rejection filters were applied to all the velocity and backscatter data while tilt filter was only applied to the current velocity direction and 'earth' components of the velocity data (i.e. eastward u, northward v and vertical w; details in technical validation section). For the other instruments, 'out-of-range' filters were based on the expected ranges, i.e. values that would be realistically impossible to attain within the study area, and/or based on default values given automatically to bad data by the logger (e.g. PAR and Qs < 0, see Table 2 for details).

6) a manual 'despiking' was finally performed by plotting the data and examining the timeseries visually. Minimal rejection was done to avoid rejecting potential 'outlier events'. As a result, some spurious data points may still be present in some timeseries.

The CTD profiles were processed using SBE data analysis software and recommended procedure as described in Donnet et al. (2018a). CTD profiles were averaged in 1 m bins and visually checked individually.





**Table 2: 'out-of-range' filters used to quality control the data.**

| Instrument | Criteria |
| --- | --- |
| ADCP | Speed<0 m/s<br>PG<25%<br>Tilt>15º<br>Surface rejection (8-30% range) |
| CTD | Depth < 0.5 m or > 250 m<br>Temperature < -2ºC or > 25ºC<br>Salinity < 5 or > 37 |
| UTBI | Temperature < -2ºC or > 25ºC |
| U20 | Depth < 0.5 m or > 250 m<br>Temperature < -2ºC or > 25ºC |
| Mastodon | Temperature < -2ºC or > 25ºC |
| Weather stations | Wind speed < 0.05 or > 40 m/s<br>Atmospheric pressure < 850 or > 1069 mbar<br>Temperature < -60ºC or > 60ºC<br>PAR < 0 umol/m$^2$/s<br>Qs < 0 W/m$^2$ |
| Tide gauge | Depth < 0.5 m or > 5 m<br>Temperature < -2ºC or > 25ºC |

### 310  4.4 UTBI depth calculation

Since our thermistors (UTBI) did not have embedded pressure sensors, the depth of their temperature records needed to be estimated. This calculation was done in three steps, increasing the accuracy of the estimate at each step:

1) Once a site depth was accurately determined, mean depth of each UTBI was determined using the mooring diagrams providing with the distance from sea-bottom of each instrument. Mean depth (with respect to Mean Sea

315      Level, MSL) was then determined as: site depth – height above sea bottom.

2) 'tidal depth', i.e. depth varying due to the tide alone, was determined using the results of tidal analyses made on the instruments equipped with a pressure sensor (i.e. ADCP, CTD and U20). One reference per mooring line was used, typically the instrument located the closest to the top of the mooring having the highest data coverage so that





320 the overall mooring tilt was best approximated (CTD or U20). Tidal analysis was done using the T_TIDE programs (Pawlowicz et al., 2002) and UTBI 'tidal depth' were calculated as: MSL depth + tide.

3) Finally, to take the mooring movement into account, i.e. lateral movements of the mooring line due to currents drag, an 'absolute depth' was determined using an estimate of the mooring horizontal tilt angle. Tilt angle was determined using the same depth timeseries from which a tidal analysis was performed for the previous step. A water level residual was then calculated as: measured water level (from MSL) – tide. This residual was then used to calculate 325 a mooring line tilt angle series as: tilt = acos(H/L) where H is the instrument height above sea-bottom at any given time and L is the instrument height at rest (i.e. its mean height). Using the horizontal tilt angle timeseries, UTBI series of height above sea-bottom were calculated as: H = L x cos(tilt) where L is the UTBI height above sea-bottom at rest. The 'absolute depth' series was then determined as: site depth - H + tide.

Overall, the mooring lines average vertical tilt were below 5º (2-4º with a standard deviation of order 1-2º) with maximums 330 on order 15-25º during extreme events, corresponding to mooring vertical displacements of about 5-15 m. These vertical 'knock-down' are large compared to the 1-2 m tidal range reported in the region (Donnet et al., 2018b) but relatively small in comparison to the mooring line length (about 5-10%), indicating good mooring performances.

It should be noted that these estimates of tilt and therefore instrument depths assume no other external variation of sea-level than the tides. Other factors such as storm surges or shelf waves can affect the sea-level on the order of 0.2-1 m in the region 335 (e.g. Tang et al., 1998, Thiebaut and Vennel, 2010, Han et al., 2012 and Ma et al., 2015). A preliminary inspection of our tide gauge records (not shown) indicate that residuals, i.e. water level variations not attributed to tides, of the same range was observed during our program.

## 4.5 In-situ comparisons (e.g. CTD profile vs. mooring data)

A CTD cast was performed after each mooring deployment and just before mooring recovery (see Figure 6 as example). Thus, 340 a total of 52 casts were available for in-situ checks (F3B03 and F3B04 pre-recovery casts were missed). The primary goal of those checks were to assess the performance of the moored UTBI lines (Temperature) and moored CTDs (Salinity).

Overall, a mean difference of 0.12ºC and associated mean standard deviation of 0.11ºC was found between the CTD profiles and UBTI observations and an overall mean salinity difference of 0.07 and mean standard deviation of 0.03 between the CTD profiles and moored CTD was obtained.

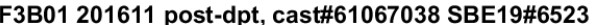

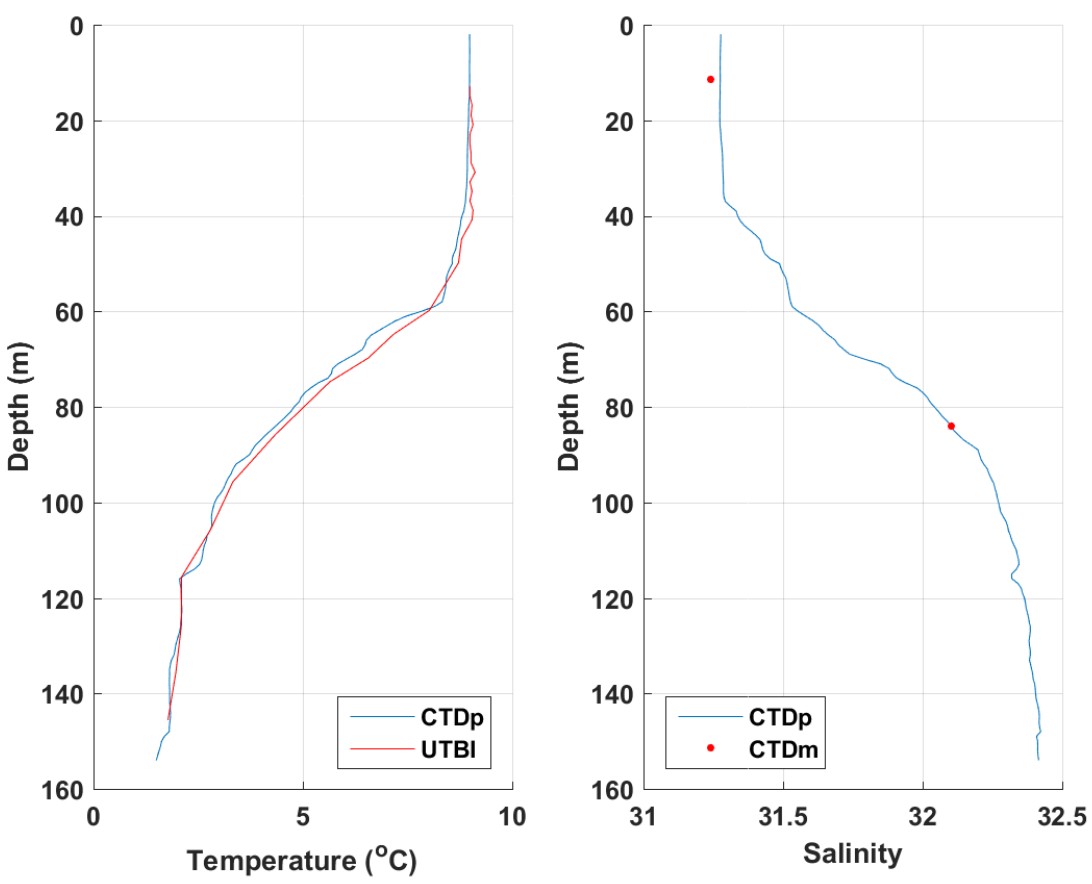

**Figure 6: in-situ CTD profile (CTDp) comparison with moored thermistors (UTBI) and CTD (CTDm) on November DD, 2016 at F3B01**

## 4.6 ADCP tilt issue

Excessive vertical tilt affects ADCPs' gyrocompass by biasing the heading which corrupt trigonometric rotation from instrument coordinates to earth coordinates (T-RDI pers. Com. and details in T-RDI documentation #P/N 951-6079-00). T-RDI indicates that while their attitude sensor can measure tilts (i.e. pitch and roll) up to about 20°, tilts above 15° will irreversibly corrupt the data (T-RDI documentation #P/N 957-6150-00, p17). If the tilt, however, stays within measurable range (i.e. 15° to about 20°) bin mapping will still hold (T-RDI pers. Com.; see details in T-RDI documentation #P/N 951-6079-00) and, thus, horizontal measurements of current speed and backscatter, i.e. variables not affected by erroneous heading, will still be correct and properly 'mapped'. At 20°, any given beam may end-up being oriented horizontally which prevents from deriving the horizontal component of the current; 3 beam solution may still work but the flow horizontal homogeneity assumption cannot be assessed (the so called 'error velocity', see T-RDI documentation #P/N 951-6069-00 p14 and P/N 957-6150-00, p14 for details) thereby limiting quality control a little bit. Beyond the tilt sensor limits (i.e. in pitch and roll axes)





which can be anywhere from 20 to about 25 degree (see Table 3), current speed calculation and bin mapping will become
biased; profiles data will then likely be unrecoverable. An illustration of this issue and potential recoverable data is presented
in Figure 7. Top 50 m (not affected by over-tilted position) and bottom (over tilted from December 7th) ADCP profiles are
plotted together, showing the effects of the tilt on current direction and current vertical component w but not on current speed
and acoustic backscatter .

In our quality control process, a 'combined' tilt angle, i.e. combination of pitch and roll angles, were used to filter unreliable
current direction and earth coordinates velocities (i.e. u, v and w). This 'combined' tilt was calculated as follow:

$$tilt = acos(cos(pitch).*cos(roll))$$

This rejection is somehow conservative since this 'combined' angle is always larger than the pitch and roll taken individually.
In addition to this automatic filter, bench tests were performed on each of our ADCPs to determine their maximum pitch and
roll angles measurable by placing each unit horizontally on a table on each direction, i.e. Beam 1-2 and Beam 3-4 axes which
helped us to further assess the quality of our data (Table 3).

Five timeseries were affected by this issue in total: F3B01 leg 2 (bottom unit), F3B01 leg 3, F3B05 leg 3&4 and F3B07 leg 3.
Being tilted near both pitch and roll limits, the latter was corrupted beyond repair and nothing could be saved from it. The
other four were generally severely tilted, but below the limits, on one side 'only'; current speed and backscatter profiles were
saved from those timeseries.

**Table 3: Severely tilted timeseries screening**

| Site and leg | Mean Pitch (º) | Mean Roll (º) | Max Pitch (º) (bench) | Max Roll (º) (bench) |
|---|---|---|---|---|
| F3B01, leg 2 | 18.1 | -0.3 | 24.2 | 23.5 |
| F3B01, leg 3 | -10.1 | 25.9 | 24.8 | 26.4 |
| F3B05, leg 3 | 22.0 | -1.1 | 24.2 | 23.5 |
| F3B07, leg 3 | 24.2 | 23.4 | 24.8 | 24.7 |
| F3B05, leg 4 | 21.5 | -0.3 | 24.2 | 23.5 |


**Figure 7: Example of an ADCP over-tilted issue. The event occurred on December 7, 2015 between 18:30 and 19:30, tilting the bottom ADCP of the mooring line on the pitch axis. Top four panels show the attitude sensor, pitch (1st panel), roll (2nd panel), calculated 'combined' tilt angle (3rd panel) and heading (4th panel and as recorded by the instrument). Red lines on the pitch and**
**tilt plots indicate the maximum sensor range as determined by bench test and the maximum accepted tilt angle of our quality control filter, respectively. The bottom four panels show current speed (mag), current direction (dir), current vertical component (w) and raw backscatter data (EA) zooming on the period 1-16 December 2015.**



## 5 Data availability

Processed data are available from the SEANOE repository (https://doi.org/10.17882/62314; Donnet and Lazure, 2020). One
file per timeseries was created in the NetCDF format containing an header with key metadata (site ID, geographic coordinates,
site depth, instrument used, author and date of creation) and the data themselves with consistent variable's naming (e.g. time,
depth, temperature etc.). UTBI timeseries were bundled together into one folder per mooring line and, thus, 1 processed file
per timeseries. CTD profiles were bundle-up per surveys and formatted as tab-delimited ASCII ODV4 files (Schlitzer, 2019).
NetCDF files were created under the MATLAB environment and tested using the NetCDF utilities (ncdump from unidata),
python (with xarray and panda libraries) and the Interactive Data Language (IDL) environments. Care was taken to export as
much data from the raw as possible (e.g. ADCP correlation magnitudes) but are provided as 'processed', that is, bad data
flagged by the QA/QC (Quality Assessment/Quality Control) process described below were replaced by NaN (Not a Number)
values.

## 6 Conclusion

We present an oceanographic dataset collected in a subpolar, mid-latitude, broad fjord. The data collection was centered on
the deployment and recovery of oceanographic moorings and a few land-based stations collecting physical parameters such
as: water temperature and salinity, ocean currents, wind speed and direction and tide. The main goal of this observation
program was to serve as a base to further study the main physical processes affecting this embayment which are likely common
to other wide stratified fjords.

To our knowledge, very few embayments alike, i.e. broad fjords, have such comprehensive observations combining numerous
and continuous in-situ sampling points. Several bays in the region have been well explored in the past but their continuous
observation via moorings rarely extended more than a few months during the spring to fall seasons, thus not offering a complete
seasonal picture (e.g. Yao 1986, de Young and Sanderson 1995, Hart et al. 1999, Schillinger et al. 2000, Tittensor et al.
2002a&b). Abroad, with the increasing interest in polar regions and their importance in climate, many recent studies relied on
significant datasets collected in broad fjords (e.g. Straneo et al. 2010, Jackson et al. 2014, Inall et al. 2015, Merrifield et al.
2018). Generally, and most likely due to extremely challenging technical constrains (e.g. massive glacial ice), those datasets
remain scarce and limited to a few points and/or few months observation, however.

By combining a relatively large number of observation points (up to 21 moorings during leg 3), high vertical resolution in both
thermal stratification (2-10 m) and ocean currents (1-3 m) as well as duration (2-years), we believe that this dataset should be
comprehensive enough to study a wide variety of processes; making it of particular interest to be shared.





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



**Appendix A**

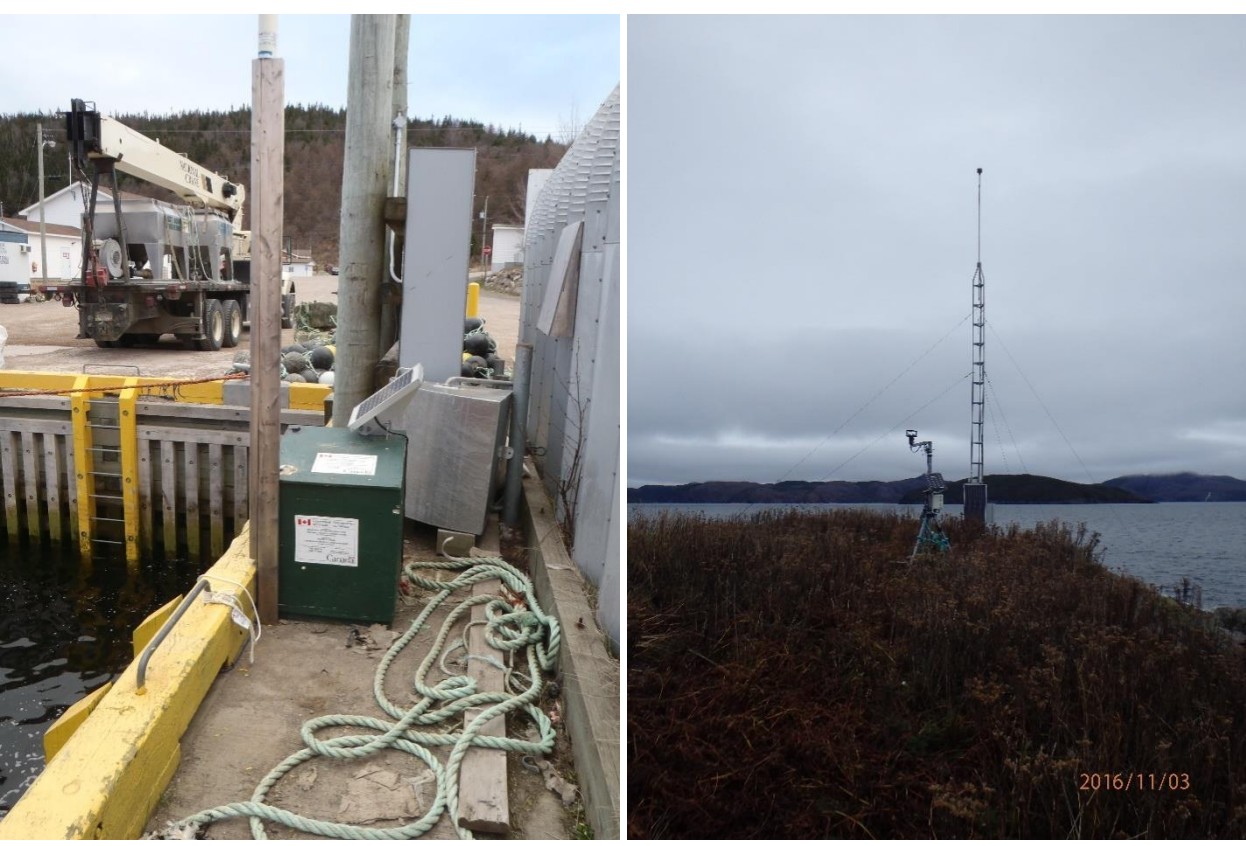

**Figure A-1: Pool's Cove (POOLC) tide gauge (left) and Dog Island (DOGIS) weather stations (right)**



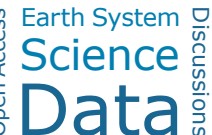

# Appendix B

**Table B-1: Data collection summary, Leg 1 (May 2015 – November 2015)**

| Site | Instrument | Latitude (ºN) | Longitude (ºW) | Deployment | Recovery | Site Depth | Instrument Depth | % Coverage |
|------|-----------|---------------|----------------|------------|----------|------------|------------------|------------|
| F3B01 | WH300#12548 | 47.193717 | -55.590867 | 2015-05-04T12:53:40 | 2015-11-14T15:33:00 | 151 | 51 | 97 |
| F3B01 | WH300#15678 | 47.193717 | -55.590867 | 2015-05-04T12:53:40 | 2015-11-14T15:33:00 | 151 | 142 | 88 |
| F3B01 | RBR#60134 | 47.194990 | -55.588950 | 2015-05-04T16:24:10 | 2015-11-14T15:47:00 | 151 | 2 | 75 |
| F3B01 | SBE19#1019 | 47.194990 | -55.588950 | 2015-05-04T16:24:10 | 2015-11-14T15:47:00 | 151 | 81 | 77 |
| F3B01 | U20#10305634 | 47.194990 | -55.588950 | 2015-05-04T16:24:10 | 2015-11-14T15:47:00 | 151 | 20 | 100 |
| F3B01 | UTBIs | 47.193717 | -55.590867 | 2015-05-04T12:53:40 | 2015-11-14T15:33:00 | 151 | 62-132 | 100 |
| F3B01 | UTBIs | 47.194990 | -55.588950 | 2015-05-04T16:24:10 | 2015-11-14T15:47:00 | 151 | 20-141 | 95 |
| | | | | | | | | |
| F3B02 | WH300#15677 | 47.342718 | -55.717833 | 2015-05-04T22:29:35 | 2015-11-14T13:07:00 | 153 | 53 | 98 |
| F3B02 | WH300#19001 | 47.342718 | -55.717833 | 2015-05-04T22:29:35 | 2015-11-14T13:07:00 | 153 | 142 | 86 |
| F3B02 | SBE19#1318 | 47.339410 | -55.923850 | 2015-05-04T19:51:20 | 2015-11-14T13:42:00 | 152 | 82 | 100 |
| F3B02 | U20#10214891 | 47.339410 | -55.923850 | 2015-05-04T19:51:20 | 2015-11-14T13:42:00 | 152 | 21 | 100 |
| F3B02 | UTBIs | 47.342718 | -55.717833 | 2015-05-04T22:29:35 | 2015-11-14T13:07:00 | 153 | 64-134 | 100 |
| F3B02 | UTBIs | 47.33941 | -55.923850 | 2015-05-04T19:51:20 | 2015-11-14T13:42:00 | 152 | 21-142 | 91 |
| | | | | | | | | |
| F3B03 | WH300#13951 | 47.557343 | -55.332225 | 2015-05-05T13:22:01 | 2015-11-17T15:17:00 | 157 | 57 | 97 |
| F3B03 | WH300#13772 | 47.557343 | -55.332225 | 2015-05-05T13:22:01 | 2015-11-17T15:17:00 | 157 | 148 | 86 |
| F3B03 | SBE37#10571 | 47.557470 | -55.329050 | 2015-05-05T15:09:07 | 2015-11-16T11:57:00 | 154 | 10 | 100 |
| F3B03 | SBE19#2245 | 47.557470 | -55.329050 | 2015-05-05T15:09:07 | 2015-11-16T11:57:00 | 154 | 81 | 100 |
| F3B03 | UTBIs | 47.557343 | -55.332225 | 2015-05-05T13:22:01 | 2015-11-17T15:17:00 | 157 | 68-138 | 100 |
| F3B03 | UTBIs | 47.557470 | -55.329050 | 2015-05-05T15:09:07 | 2015-11-16T11:57:00 | 154 | 11-142 | 96 |
| | | | | | | | | |
| F3B04 | WH300#17956 | 47.635950 | -55.281008 | 2015-05-08T12:56:00 | 2015-11-16T14:39:00 | 146 | 46 | 98 |
| F3B04 | WH300#11351 | 47.635950 | -55.281008 | 2015-05-08T12:56:00 | 2015-11-16T14:39:00 | 146 | 137 | 91 |
| F3B04 | SBE37#10572 | 47.632540 | -55.277480 | 2015-05-08T11:46:00 | 2015-11-16T14:54:00 | 260 | 6 | 100 |
| F3B04 | SBE19#2246 | 47.632540 | -55.277480 | 2015-05-08T11:46:00 | 2015-11-16T14:54:00 | 260 | 78 | 100 |
| F3B04 | UTBIs | 47.635950 | -55.281008 | 2015-05-08T12:56:00 | 2015-11-16T14:39:00 | 146 | 57-127 | 100 |
| F3B04 | UTBIs | 47.632540 | -55.277480 | 2015-05-08T11:46:00 | 2015-11-16T14:54:00 | 260 | 7-139 | 100 |





**Table B-2: Data collection summary, Leg 2 (November 2015 – May 2016)**

| Site | Instrument | Latitude (ºN) | Longitude (ºW) | Deployment | Recovery | Site Depth | Instrument Depth | % Coverage |
|------|-----------|---------------|----------------|------------|----------|-----------|------------------|-----------|
| F3B01 | WH300#12548 | 47.193783 | -55.592287 | 2015-11-15T15:47:00 | 2016-05-12T17:05:00 | 152 | 52 | 98 |
| F3B01 | WH300#15678 | 47.193783 | -55.592287 | 2015-11-15T15:47:00 | 2016-05-12T17:05:00 | 152 | 143 | 51 |
| F3B01 | RBR#22032 | 47.196250 | -55.587187 | 2015-11-15T16:00:00 | 2016-05-10T16:35:00 | 152 | 11 | 100 |
| F3B01 | SBE19#1319 | 47.196250 | -55.587187 | 2015-11-15T16:00:00 | 2016-05-10T16:35:00 | 152 | 11 | 100 |
| F3B01 | SBE19#1309 | 47.196250 | -55.587187 | 2015-11-15T16:00:00 | 2016-05-10T16:35:00 | 152 | 82 | 100 |
| F3B01 | U20#10305634 | 47.196250 | -55.587187 | 2015-11-15T16:00:00 | 2016-05-10T16:35:00 | 152 | 21 | 100 |
| F3B01 | UTBIs | 47.193783 | -55.592287 | 2015-11-15T15:47:00 | 2016-05-12T17:05:00 | 152 | 63-133 | 100 |
| F3B01 | UTBIs | 47.196250 | -55.587187 | 2015-11-15T16:00:00 | 2016-05-10T16:35:00 | 152 | 11-142 | 96 |
| F3B02 | WH300#15677 | 47.342547 | -55.717868 | 2015-11-15T13:34:00 | 2016-05-10T12:35:00 | 153 | 53 | 98 |
| F3B02 | WH300#19001 | 47.342547 | -55.717868 | 2015-11-15T13:34:00 | 2016-05-10T12:35:00 | 153 | 144 | 89 |
| F3B02 | RBR#22031 | 47.339265 | -55.717860 | 2015-11-15T12:52:00 | 2016-05-11T19:00:00 | 152 | 11 | 100 |
| F3B02 | SBE19#1317 | 47.339265 | -55.717860 | 2015-11-15T12:52:00 | 2016-05-11T19:00:00 | 152 | 11 | 100 |
| F3B02 | SBE19#1313 | 47.339265 | -55.717860 | 2015-11-15T12:52:00 | 2016-05-11T19:00:00 | 152 | 82 | 100 |
| F3B02 | U20#10305633 | 47.339265 | -55.717860 | 2015-11-15T12:52:00 | 2016-05-11T19:00:00 | 152 | 19 | 100 |
| F3B02 | U20#10214891 | 47.339265 | -55.717860 | 2015-11-15T12:52:00 | 2016-05-11T19:00:00 | 152 | 21 | 100 |
| F3B02 | UTBIs | 47.342547 | -55.717868 | 2015-11-15T13:34:00 | 2016-05-10T12:35:00 | 153 | 64-134 | 100 |
| F3B02 | UTBIs | 47.339265 | -55.717860 | 2015-11-15T12:52:00 | 2016-05-11T19:00:00 | 152 | 11-142 | 100 |
| F3B03 | WH300#13951 | 47.557268 | -55.331988 | 2015-11-17T18:06:00 | 2016-05-13T15:30:00 | 152 | 52 | 98 |
| F3B03 | WH300#13772 | 47.557268 | -55.331988 | 2015-11-17T18:06:00 | 2016-05-13T15:30:00 | 152 | 143 | 84 |
| F3B03 | SBE37#10571 | 47.558150 | -55.328900 | 2015-11-17T18:34:00 | 2016-05-13T15:58:00 | 160 | 16 | 100 |
| F3B03 | SBE19#1312 | 47.558150 | -55.328900 | 2015-11-17T18:34:00 | 2016-05-13T15:58:00 | 160 | 88 | 33 |
| F3B03 | UTBIs | 47.557268 | -55.331988 | 2015-11-17T18:06:00 | 2016-05-13T15:30:00 | 152 | 63-133 | 100 |
| F3B03 | UTBIs | 47.558150 | -55.328900 | 2015-11-17T18:34:00 | 2016-05-13T15:58:00 | 160 | 17-148 | 100 |
| F3B04 | WH300#11351 | 47.635963 | -55.281208 | 2015-11-17T11:26:00 | 2016-05-13T17:40:00 | 146 | 137 | 82 |
| F3B04 | WH300#17956 | 47.635963 | -55.281208 | 2015-11-17T11:26:00 | 2016-05-13T17:40:00 | 146 | 46 | 99 |
| F3B04 | SBE37#10572 | 47.632508 | -55.277488 | 2015-11-17T12:05:00 | 2016-05-13T17:55:00 | 260 | 5 | 100 |
| F3B04 | UTBIs | 47.635963 | -55.281208 | 2015-11-17T11:26:00 | 2016-05-13T17:40:00 | 146 | 57-127 | 100 |
| F3B04 | UTBIs | 47.632508 | -55.277488 | 2015-11-17T12:05:00 | 2016-05-13T17:55:00 | 260 | 7-138 | 100 |






**Table B-3: Data collection summary, Leg 3 (May 2016 –November 2016)**

| Site | Instrument | Latitude (ºN) | Longitude (ºW) | Deployment | Recovery | Site Depth | Instrument Depth | % Coverage |
|------|-----------|--------------|----------------|------------|----------|-----------|-----------------|-----------|
| F3B01 | WH300#12548 | 47.193460 | -55.592130 | 2016-05-13T11:11:00 | 2016-10-31T18:30:00 | 150 | 76 | 36 |
| F3B01 | RBR#22031 | 47.193460 | -55.592130 | 2016-05-13T11:11:00 | 2016-10-31T18:30:00 | 150 | 6 | 100 |
| F3B01 | SBE19#1317 | 47.193460 | -55.592130 | 2016-05-13T11:11:00 | 2016-10-31T18:30:00 | 150 | 6 | 82 |
| F3B01 | SBE19#1313 | 47.193460 | -55.592130 | 2016-05-13T11:11:00 | 2016-10-31T18:30:00 | 150 | 78 | 100 |
| F3B01 | U20#10305633 | 47.193460 | -55.592130 | 2016-05-13T11:11:00 | 2016-10-31T18:30:00 | 150 | 14 | 100 |
| F3B01 | U20#10214891 | 47.193460 | -55.592130 | 2016-05-13T11:11:00 | 2016-10-31T18:30:00 | 150 | 16 | 100 |
| F3B01 | UTBIs | 47.193460 | -55.592130 | 2016-05-13T11:11:00 | 2016-10-31T18:30:00 | 150 | 6-139 | 100 |
| F3B02 | WH300#15677 | 47.331913 | -55.737182 | 2016-05-11T16:05:00 | 2016-10-31T16:52:00 | 153 | 79 | 98 |
| F3B02 | RBR#22032 | 47.331910 | -55.737180 | 2016-05-11T16:05:00 | 2016-10-31T16:52:00 | 153 | 9 | 100 |
| F3B02 | SBE19#1319 | 47.331910 | -55.737180 | 2016-05-11T16:05:00 | 2016-10-31T16:52:00 | 153 | 9 | 65 |
| F3B02 | SBE19#1309 | 47.331910 | -55.737180 | 2016-05-11T16:05:00 | 2016-10-31T16:52:00 | 153 | 81 | 100 |
| F3B02 | U20#10305636 | 47.331910 | -55.737180 | 2016-05-11T16:05:00 | 2016-10-31T16:52:00 | 153 | 17 | 100 |
| F3B02 | U20#10305634 | 47.331910 | -55.737180 | 2016-05-11T16:05:00 | 2016-10-31T16:52:00 | 153 | 19 | 100 |
| F3B02 | UTBIs | 47.331910 | -55.737180 | 2016-05-11T16:05:00 | 2016-10-31T16:52:00 | 153 | 9-142 | 97 |
| F3B03 | WH300#13951 | 47.524650 | -55.340300 | 2016-05-16T17:00:00 | 2016-11-02T18:05:00 | 161 | 87 | 97 |
| F3B03 | SBE37#10571 | 47.524650 | -55.340330 | 2016-05-16T17:00:00 | 2016-11-02T18:05:00 | 161 | 16 | 100 |
| F3B03 | RBR#60334 | 47.524650 | -55.340330 | 2016-05-16T17:00:00 | 2016-11-02T18:05:00 | 161 | 87 | 100 |
| F3B03 | UTBIs | 47.524650 | -55.340330 | 2016-05-16T17:00:00 | 2016-11-02T18:05:00 | 161 | 17-149 | 100 |
| F3B04 | WH300#11348 | 47.639650 | -55.297100 | 2016-05-16T15:00:00 | 2016-11-04T16:15:00 | 163 | 79 | 98 |
| F3B04 | SBE37#10572 | 47.639650 | -55.297130 | 2016-05-16T15:00:00 | 2016-11-04T16:15:00 | 163 | 8 | 100 |
| F3B04 | SBE19#1315 | 47.639650 | -55.297130 | 2016-05-16T15:00:00 | 2016-11-04T16:15:00 | 163 | 80 | 100 |
| F3B04 | UTBIs | 47.639650 | -55.297130 | 2016-05-16T15:00:00 | 2016-11-04T16:15:00 | 163 | 9-141 | 100 |
| F3B05 | WH300#15678 | 47.303670 | -55.355600 | 2016-05-13T12:55:00 | 2016-11-02T16:16:00 | 91 | 80 | 38 |
| F3B05 | SBE19#1316 | 47.303670 | -55.355560 | 2016-05-13T12:55:00 | 2016-11-02T16:16:00 | 91 | 10 | 100 |
| F3B05 | SBE19#1019 | 47.303670 | -55.355560 | 2016-05-13T12:55:00 | 2016-11-02T16:16:00 | 91 | 82 | 100 |
| F3B05 | UTBIs | 47.303670 | -55.355560 | 2016-05-13T12:55:00 | 2016-11-02T16:16:00 | 91 | 10-72 | 100 |
| F3B06 | WH300#19001 | 47.437130 | -55.490800 | 2016-05-11T21:01:00 | 2016-11-02T14:50:00 | 89 | 78 | 98 |
| F3B06 | SBE19#1021 | 47.437130 | -55.490810 | 2016-05-11T21:01:00 | 2016-11-02T14:50:00 | 89 | 8 | 100 |
| F3B06 | SBE19#1237 | 47.437130 | -55.490810 | 2016-05-11T21:01:00 | 2016-11-02T14:50:00 | 89 | 80 | 100 |
| F3B06 | UTBIs | 47.437130 | -55.490810 | 2016-05-11T21:01:00 | 2016-11-02T14:50:00 | 89 | 8-70 | 100 |
| F3B07 | WH300#13772 | 47.601870 | -55.386500 | 2016-05-15T18:20:00 | 2016-11-03T14:30:00 | 96 | 85 | 0 |
| F3B07 | SBE19#1483 | 47.601870 | -55.386480 | 2016-05-15T18:20:00 | 2016-11-03T17:00:00 | 96 | 15 | 100 |
| F3B07 | SBE19#1312 | 47.601870 | -55.386480 | 2016-05-15T18:20:00 | 2016-11-03T14:30:00 | 96 | 87 | 33 |
| F3B07 | UTBIs | 47.601870 | -55.386480 | 2016-05-15T18:20:00 | 2016-11-03T14:30:00 | 96 | 15-77 | 96 |





| F3B08 | WH300#17956 | 47.580850 | -55.168450 | 2016-06-14T13:09:00 | 2016-11-05T16:24:00 | 209 | 90 | 98 |
|---|---|---|---|---|---|---|---|---|
| F3B08 | SBE19#1318 | 47.580833 | -55.168433 | 2016-06-14T13:09:00 | 2016-11-05T16:24:00 | 209 | 19 | 100 |
| F3B08 | RBR#60335 | 47.580833 | -55.168433 | 2016-06-14T13:09:00 | 2016-11-05T16:24:00 | 209 | 91 | 100 |
| F3B08 | UTBIs | 47.580833 | -55.168433 | 2016-06-14T13:09:00 | 2016-11-05T16:24:00 | 209 | 20-82 | 100 |
| F3B09 | WH600#12391 | 47.633100 | -55.440300 | 2016-05-14T16:56:00 | 2016-11-03T17:47:00 | 58 | 6 | 48 |
| F3B09 | SBE37#14435 | 47.633100 | -55.440280 | 2016-05-14T16:56:00 | 2016-11-03T17:47:00 | 58 | 7 | 100 |
| F3B09 | SBE37#14436 | 47.633100 | -55.440280 | 2016-05-14T16:56:00 | 2016-11-03T17:47:00 | 58 | 53 | 100 |
| F3B09 | UTBIs | 47.633100 | -55.440280 | 2016-05-14T16:56:00 | 2016-11-03T17:47:00 | 58 | 8-43 | 100 |
| F3B10 | WH600#12390 | 47.704750 | -55.384100 | 2016-05-14T17:39:00 | 2016-11-03T19:55:00 | 58 | 6 | 17 |
| F3B10 | SBE37#14434 | 47.704750 | -55.384060 | 2016-05-14T17:39:00 | 2016-11-03T19:55:00 | 58 | 7 | 100 |
| F3B10 | SBE37#14433 | 47.704750 | -55.384060 | 2016-05-14T17:39:00 | 2016-11-03T19:55:00 | 58 | 53 | 97 |
| F3B10 | UTBIs | 47.704750 | -55.384060 | 2016-05-14T17:39:00 | 2016-11-03T19:55:00 | 58 | 8-43 | 100 |

**Table B-4: Mooring data collection summary, Leg 4 (November 2016 – May 2017)**

| Site | Instrument | Latitude (ºN) | Longitude (ºW) | Deployment | Recovery | Site Depth | Instrument Depth | % Coverage |
|---|---|---|---|---|---|---|---|---|
| F3B01 | WH300#12548 | 47.194520 | -55.596913 | 2016-11-02T11:24:00 | 2017-05-08T17:40:00 | 156 | 82 | 99 |
| F3B01 | RBR#22031 | 47.194520 | -55.596913 | 2016-11-02T11:24:00 | 2017-05-08T17:40:00 | 156 | 12 | 95 |
| F3B01 | SBE19#1317 | 47.194520 | -55.596913 | 2016-11-02T11:24:00 | 2017-05-08T17:40:00 | 156 | 12 | 100 |
| F3B01 | SBE19#1313 | 47.194520 | -55.596913 | 2016-11-02T11:24:00 | 2017-05-08T17:40:00 | 156 | 84 | 100 |
| F3B01 | U20#10214891 | 47.194520 | -55.596913 | 2016-11-02T11:24:00 | 2017-05-08T17:40:00 | 156 | 22 | 100 |
| F3B01 | U20#10305633 | 47.194520 | -55.596913 | 2016-11-02T11:24:00 | 2017-05-08T17:40:00 | 156 | 20 | 100 |
| F3B01 | UTBIs | 47.194520 | -55.596913 | 2016-11-02T11:24:00 | 2017-05-08T17:40:00 | 156 | 12-145 | 100 |
| F3B02 | WH300#15677 | 47.330532 | -55.735460 | 2016-11-02T12:46:00 | 2017-05-08T11:59:00 | 157 | 83 | 100 |
| F3B02 | RBR#22032 | 47.330532 | -55.735460 | 2016-11-02T12:46:00 | 2017-05-08T11:59:00 | 157 | 13 | 95 |
| F3B02 | SBE19#1319 | 47.330532 | -55.735460 | 2016-11-02T12:46:00 | 2017-05-08T11:59:00 | 157 | 13 | 100 |
| F3B02 | SBE19#1309 | 47.330532 | -55.735460 | 2016-11-02T12:46:00 | 2017-05-08T11:59:00 | 157 | 85 | 100 |
| F3B02 | U20#10305634 | 47.330532 | -55.735460 | 2016-11-02T12:46:00 | 2017-05-08T11:59:00 | 157 | 23 | 100 |
| F3B02 | U20#10305636 | 47.330532 | -55.735460 | 2016-11-02T12:46:00 | 2017-05-08T11:59:00 | 157 | 21 | 100 |
| F3B02 | UTBIs | 47.330532 | -55.735460 | 2016-11-02T12:46:00 | 2017-05-08T11:59:00 | 157 | 13-146 | 97 |
| F3B03 | WH300#13951 | 47.523800 | -55.340675 | 2016-11-03T13:38:00 | 2017-05-09T13:05:00 | 159 | 85 | 99 |
| F3B03 | SBE37#10571 | 47.523800 | -55.340675 | 2016-11-03T13:38:00 | 2017-05-09T13:30:00 | 159 | 15 | 100 |
| F3B03 | RBR#60334 | 47.523800 | -55.340675 | 2016-11-03T13:38:00 | 2017-05-09T13:30:00 | 159 | 86 | 100 |
| F3B03 | UTBIs | 47.523800 | -55.340675 | 2016-11-03T13:38:00 | 2017-05-09T13:30:00 | 159 | 15-147 | 100 |
| F3B04 | WH300#11348 | 47.639702 | -55.297115 | 2016-11-05T14:55:00 | 2017-05-10T11:07:00 | 160 | 76 | 100 |
| F3B04 | SBE37#10572 | 47.639702 | -55.297115 | 2016-11-05T14:55:00 | 2017-05-10T11:07:00 | 160 | 5 | 100 |
| F3B04 | SBE19#1315 | 47.639702 | -55.297115 | 2016-11-05T14:55:00 | 2017-05-10T11:07:00 | 160 | 77 | 100 |





| F3B04 | UTBIs | 47.639702 | -55.297115 | 2016-11-05T14:55:00 | 2017-05-10T11:07:00 | 160 | 6-138 | 100 |
| F3B05 | WH300#15678 | 47.303605 | -55.357987 | 2016-11-03T11:54:00 | 2017-05-08T15:36:00 | 96 | 85 | 39 |
| F3B05 | SBE19#1316 | 47.303605 | -55.357987 | 2016-11-03T11:54:00 | 2017-05-08T15:36:00 | 96 | 15 | 100 |
| F3B05 | SBE19#1019 | 47.303605 | -55.357987 | 2016-11-03T11:54:00 | 2017-05-08T15:36:00 | 96 | 87 | 100 |
| F3B05 | UTBIs | 47.303605 | -55.357987 | 2016-11-03T11:54:00 | 2017-05-08T15:36:00 | 96 | 15-77 | 100 |
| F3B06 | WH300#19001 | 47.434265 | -55.490907 | 2016-11-03T11:20:00 | 2017-05-08T13:58:00 | 96 | 85 | 99 |
| F3B06 | SBE19#1021 | 47.434265 | -55.490907 | 2016-11-03T11:20:00 | 2017-05-08T13:58:00 | 96 | 15 | 100 |
| F3B06 | SBE19#1237 | 47.434265 | -55.490907 | 2016-11-03T11:20:00 | 2017-05-08T13:58:00 | 96 | 87 | 100 |
| F3B06 | UTBIs | 47.434265 | -55.490907 | 2016-11-03T11:20:00 | 2017-05-08T13:58:00 | 96 | 15-77 | 100 |
| F3B07 | WH300#13772 | 47.602237 | -55.386593 | 2016-11-04T15:05:00 | 2017-05-10T12:26:00 | 104 | 93 | 95 |
| F3B07 | SBE19#1483 | 47.602237 | -55.386593 | 2016-11-04T15:05:00 | 2017-05-10T12:26:00 | 104 | 23 | 99 |
| F3B07 | SBE19#1312 | 47.602237 | -55.386593 | 2016-11-04T15:05:00 | 2017-05-10T12:26:00 | 104 | 95 | 33 |
| F3B07 | UTBIs | 47.602237 | -55.386593 | 2016-11-04T15:05:00 | 2017-05-10T12:26:00 | 104 | 23-85 | 100 |
| F3B08 | WH300#17956 | 47.579797 | -55.168053 | 2016-11-06T15:00:00 | 2017-05-09T16:30:00 | 206 | 87 | 100 |
| F3B08 | SBE19#1318 | 47.579797 | -55.168053 | 2016-11-06T15:00:00 | 2017-05-09T16:30:00 | 206 | 16 | 100 |
| F3B08 | RBR#60335 | 47.579797 | -55.168053 | 2016-11-06T15:00:00 | 2017-05-09T16:30:00 | 206 | 88 | 100 |
| F3B08 | UTBIs | 47.579797 | -55.168053 | 2016-11-06T15:00:00 | 2017-05-09T16:30:00 | 206 | 17-79 | 100 |
| F3B11 | WH300#11351 | 47.478673 | -55.165868 | 2016-11-05T17:25:00 | 2017-05-09T15:13:00 | 98 | 87 | 99 |
| F3B11 | SBE37#14433 | 47.478673 | -55.165868 | 2016-11-05T17:25:00 | 2017-05-09T15:13:00 | 98 | 16 | 100 |
| F3B11 | SBE37#14434 | 47.478673 | -55.165868 | 2016-11-05T17:25:00 | 2017-05-09T15:13:00 | 98 | 88 | 100 |
| F3B11 | UTBIs | 47.478673 | -55.165868 | 2016-11-05T17:25:00 | 2017-05-09T15:13:00 | 98 | 17-79 | 100 |
| F3B12 | WH1200#13990 | 47.712317 | -55.416800 | 2016-11-04T13:22:00 | 2017-05-11T10:43:00 | 10 | 8 | 112 |
| F3B12 | SBE37#14436 | 47.713567 | -55.418317 | 2016-11-05T12:00:00 | 2017-05-11T11:05:00 | 8 | 3 | 100 |
| F3B12 | SBE37#14435 | 47.712317 | -55.416800 | 2016-11-04T13:22:00 | 2017-05-11T10:43:00 | 10 | 8 | 100 |

**Table B-5: Land-based data collection summary along the whole program (May 2015 – May 2017)**

| Site | Instrument | Latitude (ºN) | Longitude (ºW) | Deployment | Recovery | Site Depth | Instrument Depth | % Coverage |
|---|---|---|---|---|---|---|---|---|
| DOGIS | SUTRON#1204160 | 47.61463 | -55.35221 | 2015-05-04T16:50:00 | 2015-10-15T16:25:00 | -10 | -20 | 92 |
| DOGIS | SUTRON#1204160 | 47.61463 | -55.35221 | 2015-10-15T16:30:00 | 2015-11-19T17:00:00 | -10 | -20 | 100 |
| DOGIS | SUTRON#1204160 | 47.61463 | -55.35221 | 2015-11-19T17:30:00 | 2016-05-14T13:30:00 | -10 | -20 | 98 |
| DOGIS | SUTRON#1204160 | 47.61463 | -55.35221 | 2016-05-14T13:50:00 | 2016-11-03T12:30:00 | -10 | -20 | 99 |
| DOGIS | SUTRON#1204160 | 47.61463 | -55.35221 | 2016-11-03T15:20:00 | 2017-01-19T12:30:00 | -10 | -20 | 56 |
| DOGIS | SUTRON#1204160 | 47.61463 | -55.35221 | 2017-01-19T13:40:00 | 2017-05-09T18:00:00 | -10 | -20 | 98 |
| | | | | | | | | |
| DOGIS | U30#10072354 | 47.61463 | -55.35221 | 2015-05-03T19:00:00 | 2015-11-19T18:05:00 | -10 | -12 | 96 |
| DOGIS | U30#10072354 | 47.61463 | -55.35221 | 2015-11-19T18:30:00 | 2016-05-14T12:00:00 | -10 | -12 | 83 |





| DOGIS | U30#10072354 | 47.61463 | -55.35221 | 2016-05-14T12:30:00 | 2016-06-14T15:40:00 | -10 | -12 | 100 |
| DOGIS | U30#10072354 | 47.61463 | -55.35221 | 2016-06-14T16:00:00 | 2016-11-03T12:30:00 | -10 | -12 | 87 |
| DOGIS | U30#10072354 | 47.61463 | -55.35221 | 2016-11-03T15:50:00 | 2017-01-19T14:30:00 | -10 | -12 | 99 |
| DOGIS | U30#10072354 | 47.61463 | -55.35221 | 2017-01-19T14:40:00 | 2017-05-09T18:00:00 | -10 | -12 | 99 |
| | | | | | | | | |
| DOGIS | U20#10305631 | 47.61463 | -55.35221 | 2015-05-03T19:00:00 | 2015-11-19T18:05:00 | -10 | -12 | 100 |
| DOGIS | U20#10305631 | 47.61463 | -55.35221 | 2015-11-19T18:19:00 | 2016-05-14T14:00:00 | -10 | -12 | 100 |
| DOGIS | U20#10305631 | 47.61463 | -55.35221 | 2016-05-14T15:00:00 | 2016-11-03T12:30:00 | -10 | -12 | 100 |
| DOGIS | U20#10305631 | 47.61463 | -55.35221 | 2016-11-03T16:15:00 | 2017-05-09T18:30:00 | -10 | -12 | 100 |
| | | | | | | | | |
| POOLC | SUTRON#1112700 | 47.67993 | -55.43002 | 2015-05-01T18:40:00 | 2015-11-19T22:00:00 | 3 | 2 | 100 |
| POOLC | SUTRON#1112700 | 47.67993 | -55.43002 | 2015-11-19T22:10:00 | 2016-05-15T12:00:00 | 3 | 2 | 73 |
| POOLC | SUTRON#1112700 | 47.67993 | -55.43002 | 2016-05-15T13:10:00 | 2016-11-04T10:48:24 | 3 | 2 | 99 |
| POOLC | SUTRON#1112700 | 47.67993 | -55.43002 | 2016-11-04T11:00:00 | 2017-05-11T14:00:00 | 3 | 2 | 54 |


**Table B-6: IFREMER data collection summary**

| Site | Instrument | Latitude (ºN) | Longitude (ºW) | Deployment | Recovery | Site Depth | Instrument Depth | % Coverage |
|------|-----------|---------------|----------------|------------|----------|------------|------------------|------------|
| SPMGF | WH300 | 46.9581 | -56.2293 | 2015-06-22T13:11 | 2015-09-15T16:51 | 78 | 77.5 | 98 |
| SPMGF | WH300 | 46.9581 | -56.2293 | 2015-09-18T13:08 | 2015-11-18T18:58 | 81 | 80.5 | 86 |
| SPMGF | WH300 | 46.9581 | -56.2293 | 2016-03-18T13:04 | 2016-05-30T17:54 | 81 | 80.5 | 87 |
| SPMGF | WH300 | 46.9581 | -56.2293 | 2016-06-02T17:04 | 2016-11-15T16:24 | 79 | 78.5 | 98 |
| | | | | | | | | |
| M01 | MASTODON#03070 | 46.9880 | -55.9900 | 2016-05-22T12:00 | 2016-10-30T00:00 | 60 | 60 | 100 |
| M02 | MASTODON#03081 | 47.1780 | -56.1390 | 2016-05-22T12:00 | 2016-10-30T00:00 | 60 | 60 | 100 |
| M03 | MASTODON#03089 | 47.0700 | -55.8800 | 2016-05-22T12:00 | 2016-10-30T00:00 | 60 | 60 | 100 |
| M05 | MASTODON#03066 | 47.1250 | -55.770 | 2016-05-22T12:00 | 2016-10-30T00:00 | 60 | 60 | 100 |
| M06 | MASTODON#03062 | 47.2400 | -55.870 | 2016-05-22T12:00 | 2016-10-30T00:00 | 60 | 60 | 100 |
| M07 | MASTODON#03077 | 46.8250 | -56.1667 | 2016-07-13T20:00 | 2016-10-30T00:00 | 60 | 60 | 100 |
| M08 | MASTODON#03041 | 46.8667 | -56.1833 | 2016-07-13T20:00 | 2016-10-30T00:00 | 60 | 60 | 100 |
| M09 | MASTODON#03051 | 46.9500 | -56.1833 | 2016-07-13T20:00 | 2016-10-30T00:00 | 60 | 60 | 100 |
| M10 | MASTODON#03046 | 47.0250 | -56.1533 | 2016-07-13T20:00 | 2016-10-30T00:00 | 60 | 60 | 6 |
| M11 | MASTODON#03060 | 47.0917 | -56.1667 | 2016-07-13T20:00 | 2016-10-30T00:00 | 60 | 60 | 100 |
| M12 | MASTODON#03067 | 47.1250 | -56.2833 | 2016-07-13T20:00 | 2016-10-30T00:00 | 60 | 60 | 100 |