# Peer review of "A comprehensive oceanographic dataset of a subpolar, mid-latitude broad fjord: Fortune Bay, Newfoundland, Canada"

_Earth System Science Data, 2020_

## Referee Comment (RC1) · Anonymous Referee #1 · 31 Mar 2020

[english,12pt]article

**Review of "A comprehensive oceanographic dataset of a subpolar, mid-latitude broad fjord: Fortune Bay, Newfoundland, Canada" by Sebastien Donnet *et al.**

March 31, 2020

**General comments**

This paper presents a new dataset of oceanic physical parameters and atmospheric forcing parameters for Fortune Bay in Newfoundland, Canada. The data comes from oceanographic moorings and land-based stations spread all around the bay and spanning two full years (2015-2017). They form a comprehensive dataset for studying physical oceanographic processes occurring in Fortune Bay, which are also encountered in broad fjords at higher latitudes and are therefore of wider interest. The paper is well written and organized, and the data processing, limitations and quality control well described. The data is easily accessible on the online data repository mentioned in the paper. I therefore recommend its publication in ESSD after addressing the following minor issues.

[Figure]

**Specific comments**

1. Line 68: in the caption of Figure 2, legs are mentioned, which are defined later in the text but spread among many sentences. It would be better to define all leg periods in a single sentence in the text before referring to Figure 2.

2. Line 154: Since temperature is measured at the transducer head of the ADCP, its value does not need to be selected during instrument setup. The ES command is only used to select the salinity value.

3. The name of section 3.2 is confusing since this section describes the objectives of the program. Its content should therefore be moved either to the introduction or to the discussion sections, and this section removed.

**Technical corrections**

1. Line 36: downwelling

2. Line 157: PDBM rather than PDBW in the equation for Sv.

3. Line 187: instrument losses

4. Line 233: current speed accuracy

5. Line 346, Figure 6 caption: replace DD by the day in November 2016 when the profile was taken.

---

## Referee Comment (RC2) · Vladislav Petrusevich (Referee) · 3 May 2020

The study presents an interesting new dataset and initial analysis of physical oceanographic and meteorological data for Fortune Bay, Newfoundland, Canada from several oceanographic moorings and land-based tide gauge and weather stations. The record covers two years of deployment (2015-2016). There was proper quality assessment and quality control performed all the instruments and the acquired data. The quality control properly addressed the ADCP tilt issue that could affect the quality of ADCP acquired data. There was conducted in-situ comparisons between CTD profiles and the mooring data. ADCP velocity data was complemented by processing raw ADCP

backscatter to volume backscatter strength (VBS) using an updated procedure by Mullison (2017), which can be used for future studies of sediment transport or zooplankton dynamics. I can confirm that the paper is well written, data is well organized, quality control and limitations are properly addressed. I agree with another anonymous referee, that section 3.2 should be moved either to the introduction or to the discussion sections. The data is made available in an open-access online repository as NetCDF files, which is a commonly accepted format for the sharing of array-oriented scientific data. I would recommend its publication in ESSD.

---

## Author Comment (AC1) · 31 May 2020

Dear Editor-in-Chief,

many thanks to the reviewers for their constructive and positive reviews. Please see below our responses.

Anonymous Referee #1, Specific comments:

1. "Line 68: in the caption of Figure 2, legs are mentioned, which are defined later in the text but spread among many sentences. It would be better to define all leg periods in a single sentence in the text before referring to Figure 2.".

answer: we agree. The definition of the observation periods (so called 'legs') was moved up into the third sentence of the first paragraph of the "Material & methods" section as "field operations occurred in May and November of each year for about 10-15 days each time; delimiting 4 observation periods defined herein as 'leg': May-November 2015 (leg 1), November 2015 – May 2016 (leg 2), May-November 2016 (leg 3) and November 2016 – May 2017 (leg 4)". We have also slightly edited the second sentence of this paragraph to better define our naming of 'year 1' and 'year 2'. The text following those 2 sentences was then edited to remove redundancies.

2. "Line 154: Since temperature is measured at the transducer head of the ADCP, its value does not need to be selected during instrument setup. The ES command is only used to select the salinity value."

answer: the sentence was rewritten to clarify this as "Transducer temperature measured at transducer head and salinity value selected during instrument setup (ES command; 32 in our deployments) were used to calculate the water absorption (a) along the range (from transducer) R; thereby implying homogeneous water conditions."

3. "The name of section 3.2 is confusing since this section describes the objectives of the program. Its content should therefore be moved either to the introduction or to the discussion sections, and this section removed."

answer: we agree and moved this paragraph into the Discussion section (becoming then its first paragraph). As a result, sub-section heading "3.1" was also removed, i.e. "Result" section does not contain any sub-section anymore.

Anonymous Referee #1, Technical corrections:

1. "Line 36: downwelling": addressed

2. "Line 157: PDBM rather than PDBW in the equation for Sv": addressed

3. "Line 187: instrument losses": addressed

4. "Line 233: current speed accuracy": addressed

5. "Line 346, Figure 6 caption: replace DD by the day in November 2016 when the profile was taken": addressed

Vladislav Petrusevich (Referee):

"I agree with another anonymous referee, that section 3.2 should be moved either to the introduction or to the discussion sections."

answer: we moved this paragraph into the Discussion section (becoming then its first paragraph). As a result, sub-section heading "3.1" was also removed, i.e. "Result" section does not contain any sub-section anymore.

Additional comments:

while reviewing, we have taken the liberty to polish our Figure 2 with, hopefully clearer, graphics and reviewed our uncertainty of Sv from 5 dB to 3 dB based on our latest estimates of power uncertainty (from +-3 dB to +-1 dB). We have also added one reference in the conclusion upon discovering a recent paper relevant to this work (Hop et al., 2019) and slightly edited (corrected) another (Lazure et al. 2018).